# Isolated Limb Perfusion and Immunotherapy in the Treatment of In-Transit Melanoma Metastases: Is It a Real Synergy?

**DOI:** 10.3390/jpm14050442

**Published:** 2024-04-23

**Authors:** Marco Rastrelli, Francesco Russano, Francesco Cavallin, Paolo Del Fiore, Claudia Pacilli, Claudia Di Prata, Carlo Riccardo Rossi, Antonella Vecchiato, Luigi Dall’Olmo, Simone Mocellin

**Affiliations:** 1Soft-Tissue, Peritoneum and Melanoma Surgical Oncology Unit, Veneto Institute of Oncology IOV-IRCCS, 35128 Padova, Italy; marco.rastrelli@unipd.it (M.R.); francesco.russano@iov.veneto.it (F.R.); antonella.vecchiato@iov.veneto.it (A.V.); luigi.dallolmo@unipd.it (L.D.); simone.mocellin@unipd.it (S.M.); 2Department of Surgical, Oncological and Gastroenterological Sciences (DISCOG), University of Padua, 35128 Padova, Italy; carlor.rossi@unipd.it; 3Independent Statistician, 36020 Solagna, Italy; cescocava@libero.it; 4Department of Medicine (DIMED), School of Medicine, University of Padova, 35128 Padova, Italy; claudia.pacilli@studenti.unipd.it; 5Surgery Department, San Martino Hospital, 32100 Belluno, Italy

**Keywords:** metastases in-transit, in-transit melanoma metastases, isolated limb perfusion, ILP

## Abstract

Background: Isolated limb hyperthermic-antiblastic perfusion (ILP) was the most effective local treatment for advanced in-transit melanoma, but the advent of modern effective immunotherapy (IT), such as immune checkpoint inhibitors, has changed the treatment landscape. Methods: This study evaluated the role of the association between ILP and IT in the treatment of locally advanced unresectable melanoma, particularly in relation to modern systemic therapies. We analyzed 187 consecutive patients who were treated with ILP (melphalan or melphalan associated with TNF-alpha) for advanced melanoma at the Veneto Institute of Oncology of Padua (Italy) and the Padua University Hospital (Italy) between June 1989 and September 2021. Overall survival (OS), disease-specific survival (DSS), local disease-free survival (local DFS) and distant disease-free survival (distant DFS) were evaluated. Local toxicity was classified according to the Wieberdink scale and surgical complications according to the Clavien–Dindo classification. Response to locoregional therapy was evaluated during follow-up according to the RECIST 1.1 criteria (Response Evaluation Criteria in Solid Tumor). Results: A total of 99 patients were treated with ILP and 88 with IT + ILP. The overall response rate was 67% in both groups. At 36 months, OS was 43% in the ILP group and 61% in the ILP + IT group (*p* = 0.02); DSS was 43% in the ILP group and 64% in the ILP + IT group (*p* = 0.02); local DFS was the 37% in ILP group and 53% in the ILP + IT group (*p* = 0.04); and distant DFS was 33% in the ILP group and 35% in the ILP + IT group (*p* = 0.40). Adjusting for age and lymph node involvement, receiving ILP + IT was associated with improved OS (*p* = 0.01) and DSS (*p* = 0.007) but not local DFS (*p* = 0.13) and distant DFS (*p* = 0.21). Conclusions: Our findings confirm the synergy between ILP and IT. ILP remains a valuable loco-regional treatment option in the era of effective systemic treatments. Further studies are needed to establish the optimal combination of loco-regional and systemic treatments and address the best timing of this combination to obtain the highest local response rate.

## 1. Introduction

In-transit metastases (ITMs) affect 5–10% of all melanoma patients and are mainly located in the lower limbs (70%) [1]. In the case of a few small lesions, surgical excision is the treatment of choice, while bulky or rapidly recurrent lesions are best managed with locoregional and systemic therapies [2,3]. Before the introduction of systemic therapies, hyperthermic isolated limb perfusion (ILP) was the most effective treatment in case of multiple, bulky, recurrent in-transit disease [4,5]. Several large retrospective studies reported a median overall response rate of 70–90%, a median complete response of 30–70% and low/medium local toxicity [6,7]. This treatment is very effective, repeatable and is well tolerated even by elderly patients [8,9,10,11,12,13,14]. For many years, in our surgical center, ILP was combined with a systemic administration of low-dose interferon alpha 2b (LDI) to prolong the duration of the local progression-free survival [15,16]. A multicentric randomized phase III trial demonstrated that ILP had an excellent local response but no effect on the development of distant metastases [17]. Unpredictably, many patients with ITM develop distant disease (5-year overall survival of 30–40%), and this is currently considered a reason to immediately start a medical treatment [18]. Systemic immunotherapy (ICI) or therapy targeting BRAF/MEK mutations (TT) can improve the survival of stage III (unresectable) and stage IV melanoma patients, but no information is available for patients with in-transit metastases [19,20]. No information is available for patients with in-transit metastases (ITM) because patients with ITM (defining a clinical stage IIIc) were often not included in these studies. Currently, ITM patients immediately start a systemic treatment rather than receiving ILP, and some of them may receive a combination of locoregional and systemic treatment. In fact, the efficacy of ICI for the treatment of ITM and the best time for integration with locoregional therapies is unknown [21,22,23]. This study evaluated the interaction between ILP and IT in patients with unresectable ITM in terms of overall survival, disease-free survival and local disease-free survival. The treatment of ITM (defining a clinical stage IIIc) does not involve a traditional surgical approach for the extent of disease. Currently, there is indication for both TT and IT for stage III-IV. Reconsidering the role of ILP in this therapeutic setting is critical for new integration into treatment guidelines. This study aimed to assess whether ILP can still be a valid treatment for local control of bulky disease and act synergistically and compatibly with systemic IT.

In the literature, there are only two studies reporting a total of 38 patients treated with IT + ILP [24,25]. We report our single-center experience of 90 patients treated with ILP or without IT in relation to local disease control.

To our knowledge, this was the largest cohort of such patients who were treated with ILP and IT. Thus, unlike previous studies, the characteristics of our study allowed us, for the first time, to conduct a statistically significant study of the association between the two treatments.

## 2. Materials and Methods

### 2.1. Study Design

This was a retrospective cohort study on adjuvant treatment with ILP and ILP + IT for unresectable limb melanoma. The study was conducted in accordance with the principles of the Declaration of Helsinki, and all patients gave their consent for data collection and analysis for scientific purposes. The study was approved by the local ethical committee.

### 2.2. Patients

All patients who underwent ILP for unresectable limb melanoma between June 1989 and September 2021 at the University Hospital of Padua and the Veneto Institute of Oncology of Padua (Italy) were retrospectively evaluated for inclusion in the study. Patients were divided into two groups: those who underwent ILP alone and those who underwent ILP + IT. Immunotherapy included LDI (systemic administration of 3 million IU/day, 7 days/week) or ICI. Tumor staging was standardized using the eighth version of the American Joint Committee on Cancer (AJCC) [26]. Performance status following the clinical event (development of ITM or distant metastasis) was defined according to the Eastern Cooperative Oncology Group (ECOG) system [27]. Demographics, tumor characteristics, treatment data and follow-up information were extracted from the electronically available medical records.

### 2.3. Operative Technique

The surgical technique for ILP has been described in detail elsewhere [28]. In brief, under general anaesthesia, the main artery and vein were isolated and transversally incised. Specific catheters were inserted into the vessels and connected to an extracorporeal circuit where drugs were administered. In order to prevent systemic drug leakage, 99 m Tc-Albumin was injected into the circuit to measure the perfusate-to-systemic circulation leakage with a gamma probe placed over the heart and connected to a gamma counter for continuous monitoring. The TNF dose was 1 mg (used only for bulky melanoma patients) and the dosages of melphalan were 13 mg/L of the upper limb volume and 10 mg/L of the lower limb volume. All ILP procedures performed in our centers were hyperthermic (target temperature: 40.5 °C).

### 2.4. Assessment of Response

The local response was evaluated after ILP firstly at 6 weeks and later at 12 weekly intervals until the best response according to the RECIST criteria: complete response (CR) if all lesions disappeared; partial response (PR) in the case of a ≥30% decrease in the sum of the diameters of the target lesion; progressive disease (PD) in the case of a >25% increase in the size of any measured lesions, the appearance of new lesions or both; and stable disease (SD) if the previous criteria for CR, PR and PD were not met. The overall response rate (ORR) included patients with CR and PR. ILP-related local toxicity was assessed by the Wieberdink scale [29] and surgical complications by the Clavien–Dindo classification [30].

### 2.5. Statistical Analysis

Continuous data were summarized as median and interquartile range (IQR). Data were compared between two groups using the Mann–Whitney test, chi-square test and Fisher’s test. Overall survival (OS) was calculated from the date of ILP to the date of death or the date of the last visit. Disease-specific survival (DSS) was calculated from the date of ILP to the date of death due to melanoma or the date of the last visit. Local disease-free survival (DFS) was calculated from the date of ILP to the date of ITM/death or the date of the last visit. Distant DFS was calculated from the date of ILP to the date of distant metastasis/death or the date of the last visit. Survival curves were calculated using the Kaplan–Meier method and compared by means of the log rank test. Multivariable analyses of survival were carried out using Cox’s regression models including treatment (ILP vs. IT + ILP) and unbalanced characteristics at baseline. Effect sizes were reported as hazard ratios with a 95% confidence interval. All tests were 2-sided, and a *p*-value < 0.05 was considered statistically significant. Statistical analysis was carried out using R 4.3 (R Foundation for Statistical Computing, Vienna, Austria) [31].

## 3. Results

Overall, 187 patients fulfilled the inclusion criteria (99 treated with ILP and 88 treated with IT + ILP) and were included in the analysis. Demographics, tumor characteristics and treatment information are reported in Table 1. IT included IFN (*n* = 59), ICI (*n* = 12) or both (*n* = 17).

Information on the response to ILP was available in 165/187 patients (89%). Overall, ORR was 67%, including 48% CR. At a median follow-up of 25 months (IQR 9–62) after ILP, 113 patients were dead (107 from the disease and 6 from other causes) and were 74 alive. Overall, 29 patients had ITM, and 96 patients had distant metastases during the follow-up. OS at 3 years after ILP treatment was 43% in the ILP group and 61% in the IT + ILP group (*p* = 0.02; Figure 1). DSS at 3 years after ILP treatment was 43% in the ILP group and 64% in the IT + ILP group (*p* = 0.02; Figure 2). Local DFS at 3 years after ILP treatment was 37% in the ILP group and 53% in the IT + ILP group (*p* = 0.04; Figure 3). Distant DFS at 3 years after ILP treatment was 33% in the ILP group and 35% in the IT + ILP group (*p* = 0.40; Figure 4). Adjusting for unbalanced characteristics at baseline (age and lymph node involvement), receiving ILP + IT was associated with improved OS (hazard ratio: 0.59, 95% confidence interval: 0.30 to 0.89; *p* = 0.01) and DSS (hazard ratio: 0.56, 95% confidence interval: 0.37 to 0.85; *p* = 0.007) compared to ILP alone, while there was no statistically significant difference between the two treatments in terms of local DFS (hazard ratio: 0.74, 95% confidence interval: 0.50 to 1.09; *p* = 0.13) and distant DFS (hazard ratio: 0.79, 95% confidence interval: 0.55 to 1.14; *p* = 0.21).

## 4. Discussion

This study investigated the role of ILP with or without IT in the treatment of ITM patients. Our findings suggested that combining ILP and IT might offer some advantages over ILP alone in terms of OS, DSS and local DFS. Of note, local toxicity was in accordance with the literature [7,16,32,33,34,35,36,37], with less than one out of ten patients developing grade III or greater toxicity. At the state of the art, researchers usually include ITM patients among those with distant metastases and not as a stand-alone group. This represented a major limitation when reviewing the literature, as the response rate of ITM patients treated with only IT could not be extrapolated from most published studies. In our patients who were treated with ILP alone, treatment response and patient survival were consistent with the literature [7,10,11,12,13,16,32,33,34,35,36,37], while DFS could not be compared because such information was often unreported in previous studies. In our series, patients who were treated with ILP + IT showed better treatment response but a similar patient survival compared to the literature data on patients treated with IT [21,22,37]. We believe that the similar survival can be explained by the beneficial use of IT itself. The recent literature offered some studies on local response to IT in ITM patients. Guadagni et al. showed that hypoxic perfusion with melphalan was an effective treatment in patients with locoregional metastasis of advanced pelvic melanoma [38]. The authors also suggested to investigate whether hypoxic perfusion with melphalan could generate an immune response enhanced by IT with PD-L1 antibodies. A phase II clinical trial evaluated the treatment with isolated limb infusion with melphalan followed by ipilimumab administration in 26 patients [39]. This combination of local chemotherapy and IT resulted in a 3-year response rate of 85% and a 1-year progression-free survival of 58%, suggesting that locoregional treatment combined with IT could induce an excellent and durable response to therapy. Davies et al. retrospectively assessed 97 patients who received ILP, including 18 patients who had prior IT considered as immune checkpoint inhibition or oncolytic virotherapy [24]. Notably, the authors found that CR was worse in patients treated with ILP + IT, but the small number of such patients suggested caution in the interpretation of such a finding. Moreover, the authors acknowledged that the cohorts could have been unbalanced in terms of many factors. Holmberg et al. retrospectively evaluated 218 patients who received ILP, including 20 patients treated with ICI and ILP [25].

## 5. Conclusions

The introduction of modern IT did not impact the response and the local/systemic progression in this study, suggesting that ILP could still be a valid treatment option for such patients. However, the comparison focused on the “pre-ICI era” and “ICI era”; hence, the results referred to IT availability rather than the actual treatment effect because ICI was administered to only 24% of the patients in “ICI era”. Nonetheless, we believe that the literature data suggest that combining ILP with IT could be effective in terms of local response. The underlying mechanism is still unknown and may be related to the abscopal effect well described in melanoma patients [40]. To our knowledge, this was the largest cohort of such patients who were treated with ILP and IT. However, this study has some limitations that should be considered. First, the retrospective design precludes any causal associations. Second, the limited sample size suggests caution in the interpretation of the findings. Third, information on the immunologically mediated mechanisms of action of ILP was not available due to the retrospective data collection. As mentioned in the introduction, no information is available for ITM patients because patients with ITM (defining clinical stage IIIc) were often not included in studies using systemic immunotherapy (ICI) or BRAF/MEK mutation-targeted therapy (TT). ILP has been used for many years as a first-line treatment of ITM. In the period since the advent of new melanoma drugs, ILP has no longer been used, as new systemic therapies are the treatment of choice in advanced melanoma. Reconsidering the role of ILP in this therapeutic context is crucial for a new integration into treatment guidelines. Overall, our findings support the speculation that ITM patients may benefit from a combined treatment with ILP and IT. Further trials are needed to confirm such observations, which may lead to a more central integration of ILP into treatment guidelines for ITM from melanoma. In this perspective, two studies are currently recruiting patients with in-transit metastases, examining the combination of ILP and nivolumab (ClinicalTrials.gov: NCT03685890) and T-VEC and nivolumab, respectively (ClinicalTrials.gov: NCT04330430). The results will certainly shed light on the potential synergistic interaction between ILP + IT or ILP + TT.

## Figures and Tables

**Figure 1 jpm-14-00442-f001:**
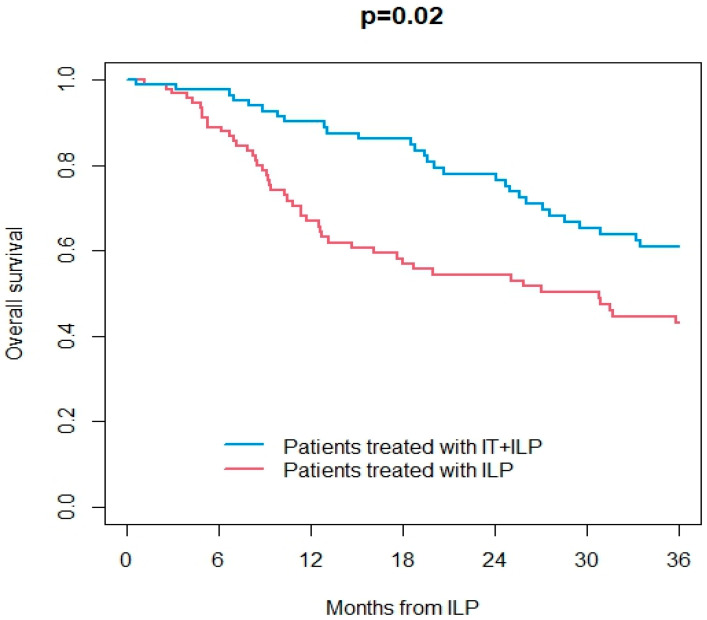
Overall survival in ILP vs. IT + ILP groups.

**Figure 2 jpm-14-00442-f002:**
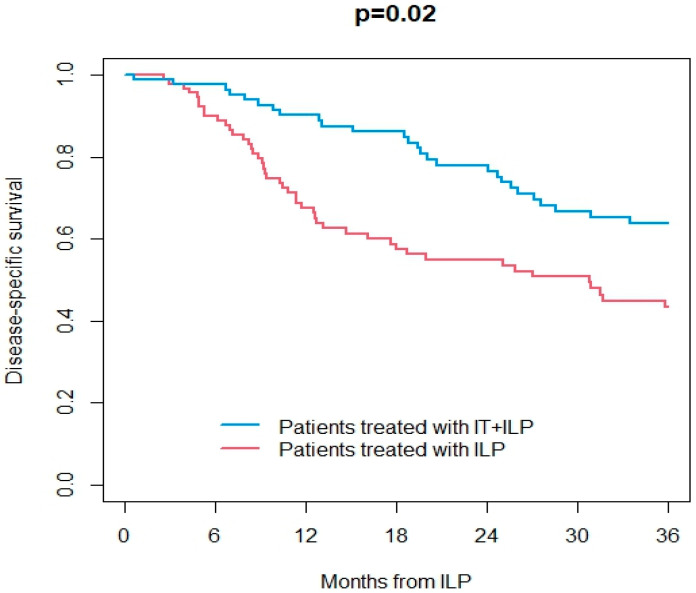
Disease-specific survival in ILP vs. IT + ILP groups.

**Figure 3 jpm-14-00442-f003:**
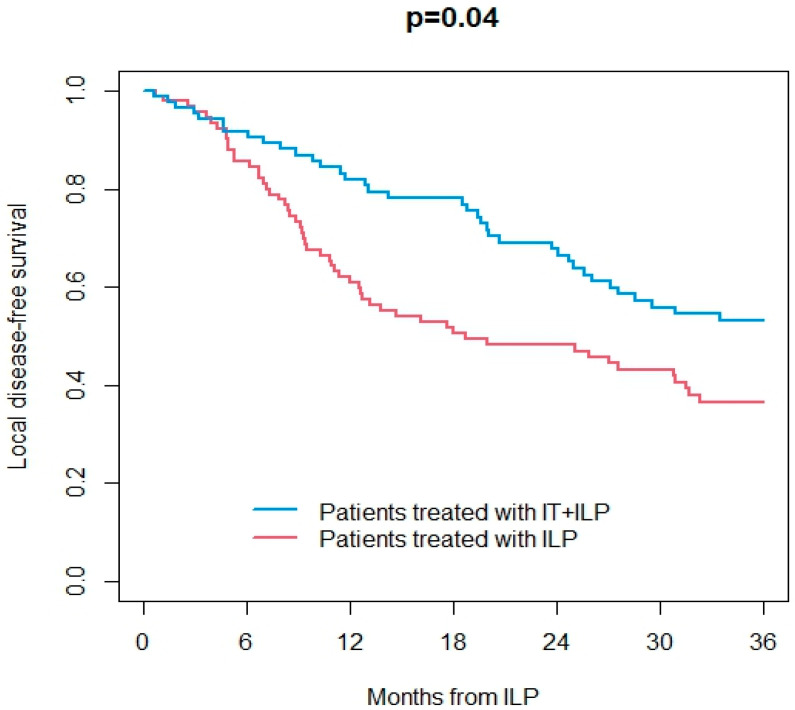
Local disease-free survival in ILP vs. IT + ILP groups.

**Figure 4 jpm-14-00442-f004:**
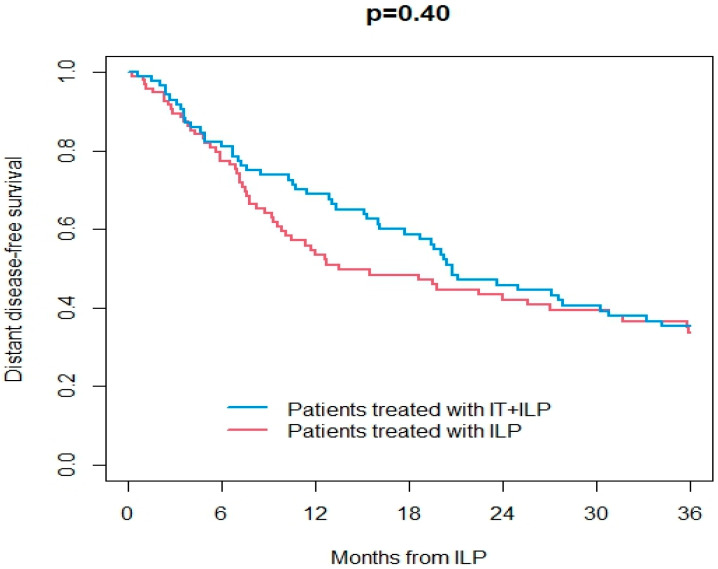
Distant disease-free survival in ILP vs. IT + ILP groups.

**Table 1 jpm-14-00442-t001:** Demographics, tumour characteristics and treatment information.

Variable	ILP (*n* = 99)	IT + ILP (*n* = 88)	*p*-Value
Males	28 (28%)	36 (41%)	0.10
Age, years ^a^	65 (54–71)	58 (49–65)	0.0004
Tumour stage:			0.89
I	6 (6%)	6 (7%)	
II	23 (23%)	18 (20%)	
III	70 (71%)	64 (73%)	
Lymph node involvement ^b^	47/98 (48%)	58 (66%)	0.02
ILP:			0.33
Melphalan	31 (31%)	21 (24%)	
TNF-Melphalan	68 (69%)	67 (76%)	
Time elapsed from primary diagnosis to ILP, months	24 (11–42)	23 (10–46)	0.92
Vascular approach:			0.83
Axillar	6 (6%)	4 (4%)	
Femoral	45 (45%)	43 (49%)	
Iliac	48 (49%)	41 (47%)	
Local toxicity (Wieberdink):			0.86
Grade I	60/98 (61%)	55 (63%)	
Grade II	30/98 (31%)	24 (27%)	
Grade III	6/98 (6%)	7 (8%)	
Grade IV	1/98 (1%)	2 (2%)	
Grade V	1/98 (1%)	0 (0%)	
Complications (Clavien–Dindo):			0.37
Grade I	95 (96%)	82 (93%)	
Grade II	3 (3%)	4 (5%)	
Grade III	0 (0%)	2 (2%)	
Grade IV	1 (1%)	0 (0%)	
Response to ILP:			0.62
CR	41/83 (50%)	39/82 (48%)	
PR	15/83 (18%)	16/82 (19%)	
PD	26/83 (31%)	23/82 (28%)	
SD	1/83 (1%)	4/82 (5%)	
ORR	56/83 (67%)	55/82 (67%)	0.99

Data summarized as n (%) or ^a^ median (IQR). ^b^ Median of three positive nodes (IQR 1–5).

## Data Availability

The datasets presented in this study can be found in online repositories. The names of the repository/repositories and accession number(s) can be found here: https://zenodo.org/records/11033673 (accessed on 16 April 2024).

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
