# Peer review of "Isolated Limb Perfusion and Immunotherapy in the Treatment of In-Transit Melanoma Metastases: Is It a Real Synergy?"

_jpm, 2024, doi:10.3390/jpm14050442_

Round 1

Reviewer 1 Report

Comments and Suggestions for Authors

The authors present an intriguing study examining the difference between treatment involving ILP and IT for locally advanced unresectable melanoma. While the study is well-documented, I have some queries that require clarification. Despite these concerns, the manuscript exhibits a well-organized structure and clear writing. Therefore, I recommend accepting the review, with the caveat that certain aspects must be addressed to enhance the paper's quality.

- The authors should consider correlating different treatments with specific mutations, particularly BRAF/MEK mutations.

- It would be beneficial to include data regarding the most prevalent mutations among the patients in Table 1.

- The authors mention the importance of BRAF/MEK mutations for treating melanomas, but there's no correlation between such mutations in the results or discussion sessions of the work presented here. 

- Enhancements to the design and colour schemes of the graphs are warranted for improved clarity and visual appeal.

Comments on the Quality of English Language

Minor editing of English language required

Author Response

SPECIFIC ANSWERS TO REVIEWERS’ COMMENTS

Reviewer #1

The authors present an intriguing study examining the difference between treatment involving ILP and IT for locally advanced unresectable melanoma. While the study is well-documented, I have some queries that require clarification. Despite these concerns, the manuscript exhibits a well-organized structure and clear writing. Therefore, I recommend accepting the review, with the caveat that certain aspects must be addressed to enhance the paper's quality. 

  1. The authors should consider correlating different treatments with specific mutations, particularly BRAF/MEK mutations.

RE: Thank you for your comment. Correlation of the different treatments to specific mutations is not possible because in our single-centre experience we selected two groups of patients treated with ILP and or without IT and therefore did not consider patients with BRAF/MEK mutations

  1. It would be beneficial to include data regarding the most prevalent mutations among the patients in Table 1.

RE: Not possible see previous explanation

  1. The authors mention the importance of BRAF/MEK mutations for treating melanomas, but there's no correlation between such mutations in the results or discussion sessions of the work presented here. 

RE: In the introduction (page 2, lanes 54-59) we claim that systemic immunotherapy (ICI) or therapy targeting BRAF/MEK mutations (TT) can improve the survival of stage III (unresectable) and stage IV melanoma patients, but we also point out that no information is available for patients with in-transit metastases citing the two most significant studies (19-20). No information is available for patients with in-transit metastases (ITM) because patients with ITM (defining a clinical stage IIIc) were often not included in these studies. ILP has been used for many years as first-line treatment of ITM. In the last period with the advent of new melanoma drugs, ILP has no longer been used, as new systemic therapies are the treatment of choice in advanced melanoma. Reconsidering the role of ILP in this therapeutic context is crucial for a new integration in treatment guidelines. This study aims to demonstrate how isolated limb perfusion is still a valid treatment for local control of bulky disease and acts synergistically and compatible with systemic immunotherapy.

  1. Enhancements to the design and colour schemes of the graphs are warranted for improved clarity and visual appeal.

RE: We thank the Reviewer for the suggestion, and we enhanced the graphs to improve clarity and visual appeal for the reader.

Sincerely,

Paolo Del Fiore

Reviewer 2 Report

Comments and Suggestions for Authors

In the manuscript by Rastrelli et al. titled “Isolated limb perfusion and immunotherapy in the treatment of in-transit melanoma metastases: is it a real synergy?”, the authors aimed to evaluate the interaction between ILP and IT in patients with unresectable ITM in terms of overall survival, disease-free survival, and local disease-free survival.

In general, the manuscript presents relevant data. The introduction clearly outlines the research objectives and current knowledge on the topic. The materials and methods section accurately describes the experimental design, and the data analysis is appropriate. Based on my assessment, with further adjustments, the manuscript could be considered for publication.

Below are some suggestions to improve the manuscript.

General comments:

As I was going through the manuscript, I noticed some minor grammatical errors and typos that could affect the overall readability of the article. Although these mistakes are minor, I suggest focusing on grammar, punctuation, sentence structure, and spelling to ensure a refined and error-free final version.

Specific comments:

Introduction section:

The authors should consider including a "significance paragraph" for this study. What sets this study apart from others? What additional value does this study offer?

Materials and methods section:

Patients:

What are the exclusion criteria? The authors should mention them if there are any.

Discussion:

The authors need to expand on their discussion section.

Comments on the Quality of English Language

English is fine. Only some minor grammatical errors and typos that should be addressed.

Author Response

Reviewer #2

In the manuscript by Rastrelli et al. titled “Isolated limb perfusion and immunotherapy in the treatment of in-transit melanoma metastases: is it a real synergy?”, the authors aimed to evaluate the interaction between ILP and IT in patients with unresectable ITM in terms of overall survival, disease-free survival, and local disease-free survival. In general, the manuscript presents relevant data. The introduction clearly outlines the research objectives and current knowledge on the topic. The materials and methods section accurately describes the experimental design, and the data analysis is appropriate. Based on my assessment, with further adjustments, the manuscript could be considered for publication.

General comments:

As I was going through the manuscript, I noticed some minor grammatical errors and typos that could affect the overall readability of the article. Although these mistakes are minor, I suggest focusing on grammar, punctuation, sentence structure, and spelling to ensure a refined and error-free final version.

RE: Done as requested 

Specific comments:

Introduction section:

The authors should consider including a "significance paragraph" for this study. What sets this study apart from others? What additional value does this study offer?

RE: We thank the Reviewer for the suggestion, and we have included a sentence in the introduction (page 2, lanes 70-76) that underlines the additional value of our study.  

Materials and methods section:

Patients:

What are the exclusion criteria? The authors should mention them if there are any.

RE: Thank you for your comment, we stratified the patients into two distinct groups: those undergoing ILP alone and those undergoing ILP+IT. We did not, therefore, consider it necessary to clarify the exclusion criteria because the inclusion criteria are strictly mandatory.

Discussion:

The authors need to expand on their discussion section.

RE: Thank you for your comment. We have added two explanatory sentences (page 7, Lines 207-210 and 217-221)

Comments on the Quality of English Language

English is fine. Only some minor grammatical errors and typos that should be address

RE: done as requested minor grammatical mistakes were encountered.

We hope the revised version is now suitable for publication and look forward to hearing

from you.

Sincerely,

Paolo Del Fiore

Round 2

Reviewer 1 Report

Comments and Suggestions for Authors

No comments. 

Reviewer 2 Report

Comments and Suggestions for Authors

I went through the revised version of the manuscript. The authors addressed all the issues raised and made the corresponding changes in the manuscript. Therefore, the manuscript can be accepted for publication in its current form.